# Co-development and piloting of a menstrual, sexual and reproductive health intervention to improve social and psychological outcomes among secondary schoolgirls in Northern Tanzania: the PASS MHW study protocol

Elialilia Okello,[1] Jennifer Rubli [ID],[2] Belen Torondel,[3] Kenneth Makata,[1] Philip Ayieko,[1,4] Saidi Kapiga,[1,4] Giulia Greco,[5] Jenny Renju [ID] [6,7]

For numbered affiliations see end of article.

**Correspondence to**
Dr Elialilia Okello;
elialilia.okello@mitu.or.tz

## ABSTRACT

**Introduction** Poor menstrual health negatively impacts psychosocial and physical health, and subsequently leads to poor school outcomes, but the effort to improve adolescent girls' menstrual health in Tanzania remains fragmented. This study aimed to develop and pilot a scalable, comprehensive menstrual, sexual and reproductive health (MSRH) intervention within Tanzanian government structures to improve MSRH practices and perceptions and the overall school climate to ensure the psychosocial well-being and optimal school participation and performance among secondary schoolgirls.

**Methods and analysis** This study will be conducted in three phases. Phase I will be a formative research to iteratively refine an existing puberty and menstrual health intervention, and to collaboratively design strategies to embed the intervention into government structures thereby promoting scalability. In Phase II, we will pilot and evaluate the refined intervention and implementation strategies using a mixed-methods design to assess (1) feasibility, acceptability and sustainability; and (2) effect on MSRH practices and perceptions and the overall school climate. In Phase III, we will synthesise the research findings in collaboration with the national, regional and district government and non-government stakeholders.

**Ethics and dissemination** This pilot study will provide evidence-based recommendations for a comprehensive, complex menstrual and puberty intervention within secondary schools in Tanzania that can be further tested for broader effectiveness across a larger population. Papers, policy briefs and both regional/international conference presentations are planned to reach academic and non-academic groups. Protocol, tools and consent have been reviewed and approved by the independent Tanzanian national ethics committee (NIMR/HQ/R.8a/Vol.IX/3647) and the LSHTM Observational/Interventions Research Ethics Committee (LSHTM Ethics Ref: 22854). The project will involve adolescents, and procedures will be followed to ensure that we obtain permission and

## Strengths and limitations of this study

► The participatory methodologies will enable meaningful stakeholders' engagement to discuss synergies between the intervention and the current school curriculum and explore mechanisms for integration in local government authority mechanisms and scaling.

► The costing component of the project will also guide scale-up strategies.

► Our inclusion of multiple outcomes from a series of different validated sources, our mixed methods and triangulation will provide important lessons as to what outcomes are best at capturing changes and to ensure they are really measuring the correct constructs.

► This is a pilot intervention and will be conducted in one region only, involving small number of schools and small number of participants.

► The study design lacks a control group and we are not randomising participants, therefore unmeasured confounding may be a major issue.

consent of parents and guardians and assent from all adolescents below 18 years of age that will be enrolled in the study.

## INTRODUCTION

Data from the 2018/2019 Tanzania education sector performance review reports low completion rates for secondary schoolgirls in Mwanza region. Of the students enrolling for secondary education in Mwanza in 2016, 67% of girls and 70.6% of boys completed the 4 years of secondary education. After truancy (the leading reason for dropout in

both groups), pregnancy accounted for 11% of dropouts among girls.[1] Yet, school completion among girls is a 'social vaccine' against multiple sexual and reproductive health (SRH) challenges such as early marriage, teen pregnancy, HIV, and child and maternal mortality. Negative menstrual experiences are associated with poor educational outcomes such as low performance and participation, school absenteeism and dropout; and poor mental and physical health, including depression, anxiety, menstrual pain, and reproductive and urinary tract infections (RTI/UTIs).[2] These issues are barriers to development and undermine progress towards most of the Sustainable Development Goals (SDG 1-4,6, 8, 10–13, 16, 17).[3]

In sub-Saharan Africa (SSA), parents, relatives and teachers are the main sources of puberty, sexual and reproductive health information for adolescents. However, these adults are often ill-informed and/or uncomfortable discussing these topics.[4 5] Further, the lack of integration of puberty and menstruation topics and SRH education leads to significant gaps in and subsequently impacts attitudes and practices.[6] A recent survey of 432 boys and 524 girls across four mixed-sex schools found that nearly half (47%) of girls reported leaving school early during their last period and nearly one-third reported lower participation (31%) and concentration (33%) in school on period days. Bullying and harassment from boys and teachers cause menstrual-related shame and anxiety.[5 6] The same study found that boys reported to 'tease' because they themselves were embarrassed, they did not know what periods were and they were also negatively influenced by their peers.[4 7–9] In addition, many schools lack adequate water, sanitation and hygiene (WASH) facilities. According to the 2018 school WASH report, availability of basic sanitation services, defined as improved usable single-sex sanitation facilities, varied greatly between rural and urban schools, with more than a half of schools in urban areas (51.0%) having basic sanitation services compared with 24.3% of schools in rural areas. Similarly, the coverage of basic hygiene services was as low as low 17.6%. Only 14.4% of government-owned schools had basic hygiene services, compared with 39% among schools owned by non-government institutions. There were also significant variations in the availability of basic hygiene services between regions, ranging from 1.4% to 47.2%.[10]

Evidence from Kenya, The Gambia and Uganda suggests that multicomponent MSRH interventions have the potential to improve health and social outcomes among adolescent schoolgirls.[11–13] Two recent trials have shown that school MSRH interventions can impact the overall school climate,[10 14] broadly defined as the perceived atmosphere of the school in regard to student commitment and engagement. In addition, broader SRH education interventions have been internationally recognised as important and effective at improving SRH knowledge and attitudes and recommended for widespread adoption.[6 15]

In Tanzania, and across much of SSA, implementation of MSRH interventions remains inadequate; small scale and patchy in coverage. Progress has been hindered by interrelated factors including (1) attitudes that such programmes are extracurricular; (2) science curricula that only include basic biological functions of reproduction and exclude practical aspects of MSRH such as life skills and sociocognitive development and anything related to sexual health, for fear of encouraging sexual activity; and (3) a lack of prioritisation, resources and time allocation within already-busy school schedules.[16]

There is growing evidence that supports the use of well-trained external facilitators to deliver MSRH education in school settings.[2 13 17] Students feel more comfortable discussing MSRH issues with adults who are not their teachers, especially in conservative contexts. Additionally, the backdrop of teacher abuse and forced pregnancy testing creates a hostile environment, not conducive to open conversations which are needed to ensure students receive the correct information.

Little is known as to how to implement and integrate acceptable, feasible, sustainable and scalable MSRH interventions into government programmes in resource-limited settings.

### Study aims and objectives
#### Aim
This study aimed to develop and pilot a scalable, comprehensive MSRH intervention within Tanzanian government structures to improve MSRH practices and perceptions and the overall school climate to ensure psychosocial well-being and optimal school participation and performance among secondary schoolgirls.

The specific objectives are as follows:
1. To refine an existing Non-Governemental Organization (NGO)-led MSRH programme to better involve boys, further develop the pain management activities and include small and sustainable school WASH facility improvements.
2. To work with a local NGO, the local government and school authorities to develop mechanisms to integrate the MSRH intervention within government school systems and structures to promote uptake, ownership and sustainability.
3. To pilot the refined intervention to assess (1) feasibility (including cost per student, school and district), acceptability and sustainability and (2) effect on MSRH practices, perceptions and the overall school climate.

### Theoretical framework
The study, named '**PA**rtnering to **S**upport **S**chools to promote good **M**enstrual **H**ealth and **W**ell-being' or PASS MHW study, has a theory of change that draws from the Integrated Model of Menstrual Experience.[18] The Integrated Model of Menstrual Experiences was developed from a qualitative meta-synthesis of 76 studies involving women and girls in low and middle income countries (LMICs) and found that social support, behavioural

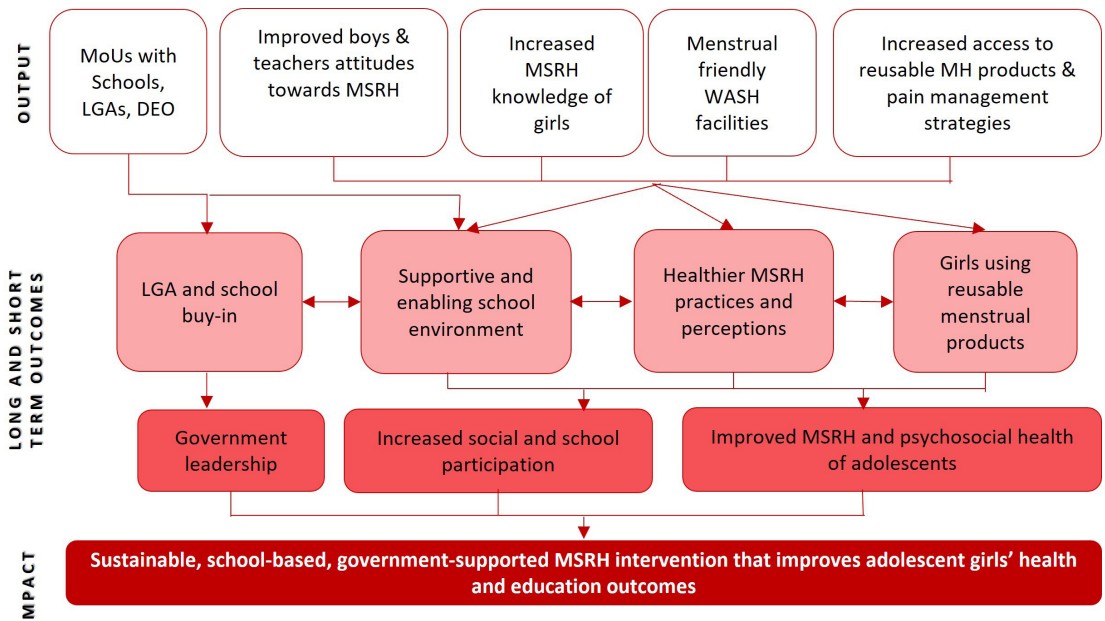

**Figure 1** Theory of change for the PASS MHW Project.

expectations, knowledge, and the physical and economic environment all affect elements of girls' menstrual experience, including practices and perceptions, confidence to participate and menstrual anxiety. In PASS MHW, we theorise that engaging local government authorities (LGAs) and other education stakeholders will foster a conducive physical and social environment, participatory MSRH education sessions will reduce gaps in MSRH knowledge and the provision of reusable menstrual products will break down economic barriers, and together with improved pain management strategies will improve social and school participation and MSRH (figure 1).

## METHODS AND ANALYSIS
### Study setting and participants
This mixed-methods study will be conducted in 10 government secondary schools in Kilimanjaro and Mwanza regions in North West Tanzania.

Kilimanjaro region, located in the northeast of mainland Tanzania, boasts some of the highest secondary school enrolment, attendance and pass rates in the whole country.[19] Mwanza region lies in the northern part of Tanzania and is among the regions with the highest failure to complete secondary school and highest rates of pregnancy-related school dropouts.[19] The Mwanza Regional Annual Education Report (2019/2020) reported that of the 53,981 students registered for secondary school, 3,753 (7%) dropped out over the course of the school year; adolescent pregnancy was listed as the third most common reason for dropout after child labour and absconding.[20]

Participants for the proposed study will include secondary school students (girls and boys) in Forms Two and Three (age range from 12 to 18 years), teachers and LGA officials, with a specific focus on those within the education offices. In order to capture maximum variation in experiences, the students and teachers' population will be drawn from rural/urban and day/boarding secondary schools.

### The TWAWEZA programme
Femme International (a registered NGO in Tanzania and Kenya and hereafter referred to as 'Femme') has been implementing the TWAWEZA (Swahili word meaning 'we can') programme in secondary schools in Kilimanjaro region for 8 years. Twaweza aims to address MSRH knowledge gaps and to provide skills and sanitary materials to adolescent girls. By the end of 2020, Femme had reached 6,440 girls in 33 schools.[21] Monitoring and evaluating data from TWAWEZA programme suggest that, in its current format the intervention is successful in improving girls' practices, menstrual health, ability to participate and confidence. However, Femme has identified various gaps in their current programme, namely, little to no involvement of boys, limited pain management options and a limited reach or scale. The aim of PASS MHW is to refine the intervention to address these gaps and then pilot the refined intervention in secondary schools in Mwanza region.

### Patient and public involvement
The research question and outcome measures were designed in partnership between the two research institutions and the implementing NGO, Femme International. As stated in the section above, Femme International had identified a gap in their programming. No patients were involved in the design of the research question and outcome measures; however, the previous implementation experiences of Femme International contributed to the design of the study. Patients were not involved in the recruitment to and conduct of the study. The results will

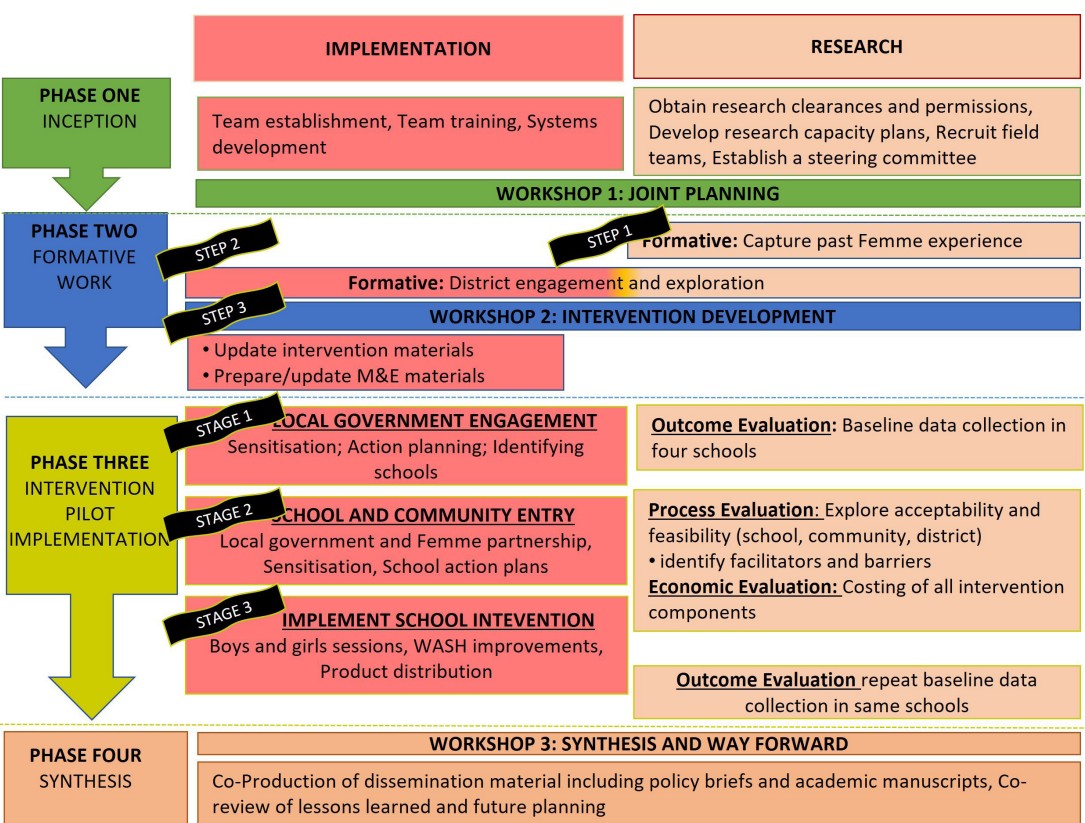

**Figure 2** Schematic representation of the different phases and stages of the PASS MHW Project.

be disseminated to the study participants through school, district and regional level meetings.

## Study procedures

The study will be implemented in three phases (see figure 2): Phase I will be a formative study to iteratively refine the Twaweza programme and together with LGAs, collaboratively design strategies to embed the intervention into government structures thereby promoting scalability. In Phase II, we will pilot and evaluate the refined intervention and implementation strategies using a mixed-methods study to assess (1) feasibility (including cost per student, school and district), acceptability and sustainability; and (2) effect on MSRH practice and perceptions and the overall school climate. In Phase III, we will synthesise the research findings in collaboration with the national and regional government LGA officials and non-government stakeholders to provide recommendations for future implementation and testing.

## Phase I: formative study

Formative research will be conducted to refine the intervention in four secondary schools in Kilimanjaro and two secondary schools in Mwanza region. This formative phase of the study will comprise three steps. The full details of all the qualitative data collection are included in table 1.

### Step 1

This step will involve collection of information about acceptability and perceived effectiveness of the current TWAWEZA programme to help refine the intervention. Questions and observations will be related to exploring which components worked or did not work well, feedback on the way in which the intervention was delivered, and to elicit participant input on specific new components that can be added to improve the intervention to: (1) involve boys and (2) improve pain management strategies. Experiences of previous participants of the TWAWEZA programme will be documented in four secondary schools in two districts in Kilimanjaro region. All schools which have received the Twaweza intervention will be stratified by year of delivery. We will only include schools that are mixed (boys and girls) and that are accessible during the rainy period. We will randomly select two schools from the list that received the intervention recently (2021) and two from those that received it previously (2019). COVID-19 restrictions in 2020 meant that no schools received the TWAWEZA intervention in 2020. Our sample will allow us to explore the short-term and long-term influences and impacts. In each of the four schools we will conduct two focus group discussions (FGDs), stratified by sex and two key informant interviews (KIIs) with school staff. We will triangulate these data with four KIIs with Femme facilitators. After data collection, we will have a participatory workshop with the Femme facilitation team to iteratively explore and discuss the initial formative phase findings.

**Table 1** Inclusion criteria, sampling strategy, aims and data collection methods

| | Participant category | Study phase and step | Data collection method | Number of participants | Sampling strategy | Targeted information |
|---|---|---|---|---|---|---|
| 1 | Students with previous experience of Femme TWAWEZA programme (Kilimanjaro region) | Phase I: Formative Step 1: Kilimanjaro | FGD in four schools, 2 per school—total 8 FGDs | ~32 girls and ~32 boys | Following school administration and parents meetings, in which we will receive consent for the study, we will present the whole study to the entire form three class and then request for the students assent to take part should they be selected. Students will be told about the recruitment strategy and that they are only eligible if they had previously taken part in the TWAWEZA programme. We will then randomly select (or recruit those randomly selected from the femme list) 8 girls and 8 boys per school. These groups will be invited to return for the data collection at a date agreed on by the school, male and female groups will be interviewed separately. | ▲ Compile learning experiences of previous participants. <br> ▲ Discuss if/how participation in the TWAWEZA programme has impacted on their lives. <br> ▲ Likes and dislikes of the programme. <br> ▲ Suggestions for the increased involvement of boys. <br> ▲ Suggestions for suitable pain management strategies (girls). <br> ▲ Discuss the product choice options (girls). |
| 2 | School staff from schools who have already received the Femme TWAWEZA programme (Kilimanjaro region) | Phase I: Formative Step 1: Kilimanjaro | KII in 4 schools with 2 KIs per school | 8 school staff | Femme will provide a list of the point people they have liaised with in each school during the implementation of the TWAWEZA intervention. This is normally the head teacher and the matron. Both staff members will be invited to take part in the study and interviews will be conducted at a time and date convenient for them. | ▲ Feedback on their experience with the TWAWEZA programme. <br> ▲ Suggestions for programme enhancements including mechanisms for the distribution of analgesics; acceptability of cognitive approaches for pain management; involvement of boys; building ownership within the school and mechanisms for working through the district government offices. |
| 3 | Femme facilitation team | Phase I: Formative Step 1: Kilimanjaro | FGD/group discussion | 4 | We will adopt a census approach and include all of the FEMME facilitators who are available for interview. | ▲ Explore their experiences of implementing the TWAWZA intervention, capture challenges and successes. <br> ▲ Elicit suggestions for programme enhancements, specifically around the pain management and the involvement of the boys. |
| 4 | Students (boys and girls) from Mwanza region | Phase I: Formative Step 2: Mwanza | FGDs in 2 schools, 2 per school—total 4 FGDs | ~16 girls and ~16 boys | Schools from two purposively selected districts in Mwanza region (Nyamagana and Misungwi) will be stratified as day and boarding. We will then randomly select one school per strata. We will conduct district level, school administration and parents meetings in 2 schools randomly selected. We will then meet with all of Form X and describe the PASS Project, those that consent to take part will then be eligible for recruitment. We will randomly select 8 boys and 8 girls using a lottery (draw from a hat) system. These groups will be invited to return for the data collection at a date agreed on by the school, male and female groups will be interviewed separately. | ▲ Assess MSRH needs and priorities. <br> ▲ Explore the current WASH situation in the school from the students perspective. |
| 5 | National, regional and district educational authorities | Phase I: Formative Step 2: Mwanza | Observation of Femme-led meeting | ~16 participants+4 Femme facilitators | No sampling as Femme will convene the meeting, the research team will be introduced at the start and state that they are there to observe the process of the meeting. <br> No recordings or names will be taken. <br> All workshop participants will be asked to consent to the observation. | ▲ Capture the perspectives of the participants regarding their ideas for synergies between the intervention and the current school curriculum. <br> ▲ Understand their perceived needs, preferences and priorities for MSRH education in schools. |
| 6 | School administrators | Phase 1: Formative Step 2: Mwanza | Observations of Femme/District led meetings | ~10, 5 per school | | ▲ Capture discussions around the proposed WASH and pain management components, particularly possible challenges and opposition. <br> ▲ Using PIPA we will capture the engagement strategies applies and adopted and any successes and challenges faced. |

Continued

**Table 1** Continued

| | Participant category | Study phase and step | Data collection method | Number of participants | Sampling strategy | Targeted information |
|---|---|---|---|---|---|---|
| 7 | Femme facilitation team and district officials | Phase I: Formative Step 3: Mwanza | Observations of meeting | 30 | ▶ We will adopt a census approach and include all of the FEMME facilitators who lead the meeting and all district officials who attend. | ▶ Elicit suggestions for programme enhancements, specifically around the pain management and the involvement of the boys and WASH improvements. |
| 8 | Female and male students | Phase II: Pilot study, Mwanza | Quantitative assisted, self-completed paper-based survey conducted among a longitudinal cohort before implementation of the intervention and then 9–12 months after | ~500 girls and 200 boys | ▶ All girls in Forms 2 and 3 in each school. ▶ Random sample of 50 boys per school, randomly selected from forms 2 and 3 using the school registers. | ▶ MSRH knowledge<br>▶ WASH facilities<br>▶ Menstrual needs and practices (girls only)<br>▶ SRH symptoms (girls only)<br>▶ Generalised anxiety<br>▶ School environment |
| 9 | Staff in pilot schools (n=4) | Phase II: Pilot study, Mwanza | KIIs conducted at month 1 and month 7 post intervention start | 12 (at two time points) | ▶ During our meeting with all the school administration, we will explain how we would like to get more detailed information from three teachers per school and explain which. ▶ The head teacher, matron and the teacher appointed to life skills sessions will be purposively recruited. ▶ The three teachers will be provided with an information sheet and consent forms and suitable times and locations will be arranged for the interviews. | ▶ Acceptability of a school-based MSRH intervention.<br>▶ MSRH knowledge and awareness of girls' and boys' needs.<br>▶ WASH facilities — opportunities for management and challenges.<br>▶ At month 7, we will also assess changes, discuss any challenges faced and explore ways to overcome any challenges. |
| 10 | District officials from pilot districts (n=2) | Phase II: Pilot study, Mwanza | KIIs conducted at month 1 and month 7 post intervention start | 4 (at two time points) | ▶ During our planning meeting in the district education offices, specifically the district education offices. We will introduce our evaluation plan and our wish to conduct sequential KIIs with two officials per district. We will purposively select two officials who plan to be/are most involved with school health and the PASS project. ▶ We will provide more detailed study information and consent forms and suitable times and locations will be arranged for the interviews. | ▶ Acceptability of a school-based MSRH intervention.<br>▶ MSRH knowledge and awareness of girls' and boys' needs.<br>▶ WASH facilities — opportunities for management and challenges.<br>▶ At month 7, we will also assess changes, discuss any challenges faced and explore ways to overcome any challenges.<br>▶ Explore the type of engagement that the district have with other stakeholders and how they view this. |
| 11 | School and district authorities from 4 schools and 2 districts | Phase II: Pilot study, Mwanza | Meeting observations | 20–35 | ▶ The external evaluation team (MITU) will attend and observe any Femme-District meetings. ▶ The observer will introduce themselves as part of the external process evaluation team. ▶ No names will be documented on the observation forms. | ▶ Engagement strategies — how Femme engages with the District.<br>▶ Opportunities and barriers for uptake and project ownership.<br>▶ Synergies between the intervention and the current school curriculum.<br>▶ Acceptability of the intervention. |
| 12 | Students in pilot schools | Phase II: Pilot study, Mwanza | Observations of Femme teaching sessions (n=10 with girls, n=8 with boys) | 160 girls and 160 boys (based on ~20 per class), | ▶ We aim to observe 4 first sessions, 4 second sessions and 2 8-week check sessions for girls, we will then observe 2 boys sessions per school. ▶ We will randomly select 2 of the schools to observe the 6–8week check in sessions. ▶ The observer will introduce themselves as part of the external process evaluation team. ▶ No names will be documented on the observation forms. | ▶ MSRH knowledge<br>▶ Engagement/participation in the sessions<br>▶ Fidelity to the intervention intended contents |

Continued

**Table 1** Continued

| | Participant category | Study phase and step | Data collection method | Number of participants | Sampling strategy | Targeted information |
|---|---|---|---|---|---|---|
| 13 | Teachers in pilot schools | Phase II: Pilot study, Mwanza | Observation of Femme sessions with teachers | 20 teachers (5 per school) | ▲ We will randomly select 2 of the schools to observe the teacher sessions.<br>▲ The observer will introduce themselves as part of the external process evaluation team.<br>▲ No names will be documented on the observation forms | ▲ Acceptability<br>▲ Fidelity<br>▲ Engagement and participation in the sessions |
| 14 | Parents of students in pilot schools | Phase II: Pilot study, Mwanza | Observation of Femme— school-led parents meetings | ~200–500 | ▲ The external evaluation team (MITU) will attend and observe any Femme-school meetings—1 in each school.<br>▲ The observer will introduce themselves as part of the external process evaluation team.<br>▲ No names will be documented on the observation forms. | **Acceptability of the intervention**<br>▲ Engagement strategies — how Femme engages with the parents and with the school.<br>▲ Opportunities and barriers for uptake and school project ownership. |
| 15 | Students in pilot schools | Phase II: Pilot study, Mwanza | Repeated IDIs | 20 girls (5 from each school) | ▲ During the full class meeting when we introduce the study, we explain how we are wanting to conduct additional data collection (as well as the questionnaire) with a smaller sample of girls to better document their experiences.<br>▲ We will randomly select a subsample of girls who reported in the baseline questionnaire to have missed two or more schools days during and because of their last menstrual period<br>▲ We will invite these girls into a IDI (without disclosing the selection process, they will be able to do this themselves if they are comfortable)<br>▲ The IDIs will take place post baseline and endline survey | ▲ Ability to manage pain<br>▲ Self-efficacy in the management of menstruation<br>▲ School and social participation |
| 16 | School level WASH observation | Phase II: Pilot study, Mwanza | WASH observation checklist | 4 schools | ▲ All schools<br>▲ Checklists will be completed immediately post intervention, after 6 months and again at 12 months | ▲ Document any improvements made<br>▲ Document changes over time |

FGD, focus group discussion; MSRH, menstrual, sexual and reproductive health; SRH, sexual and reproductive health; WASH, water, sanitation and hygiene.

## Step 2

We will conduct an assessment of MSRH needs, identify synergies between the Twaweza intervention and the current school curriculum, explore mechanisms for integration with LGA educational officers and teachers, assess WASH facilities and collaboratively agree on basic WASH improvements. This step will be conducted in one urban district in Mwanza, where we will randomly select two mixed secondary schools that have not previously received any MSRH intervention. We will conduct four FGDs with boys and girls (separately), two school-level meetings with the teachers of health-related studies and the school administration and two WASH structured observations.

In addition, we will pilot test the quantitative survey tool that will be used later to evaluate the impact of the intervention during Phase II. The draft quantitative survey tool has nine sections, which are as follows: socio-demographic information, school WASH facilities, knowledge of puberty and menstruation, menstrual experience, menstrual and reproductive health symptoms, self-efficacy, menstrual pain, school participation and school climate and draws on questions from validated[22] and non-validated survey instruments. The pilot will involve groups of 10–20 female students and 10–20 male students per school to assess students' understanding of the questions. After the assisted self-completion of the questionnaire, the facilitator will guide a discussion with a small group of selected students to assess comprehension of questions, the answer options and identify any points of confusion. Any suggested modifications will be recorded and discussed with the project team. Modifications will be made; an additional pilot (following the same steps) may be required prior to completion.

## Step 3

As part of formative research, we will hold one participatory meeting with government stakeholders from the national and regional levels, the two participating districts in Mwanza (where the pilot is taking place), and school stakeholders including headmasters, matrons, as well as any teachers who wish to attend. Data from the formative study will inform the intervention components, the evaluation framework and the data collection tools.

## Phase II: pilot implementation of the intervention
### Sampling strategy

The District Education Officers (DEOs) will provide a list of mixed-sex secondary schools from two districts in Mwanza region (as agreed on in the formative phase). A shortlisting will be conducted based on student enrolment (targeting mid-sized schools enrolling between 125 and 140 girls in Forms 2 and 3), location (to include both urban and peri-urban schools), proximity to Mwanza city, and both day and boarding mixed-sex schools where possible. A random sample of four schools will be obtained from the shortlisted schools.

In each school, all girls in Forms 2 and 3 who assent to participate and a random sample of 50 male students enrolled in the same classes as the girls, will be invited to join a longitudinal cohort study, within which we will administer a pre and post intervention assisted self-completed paper-based survey (as described above). We determined that 500 girls—each participating in the baseline and end line survey—were required to provide 80% power to detect a difference of 0.075 points in the Menstrual Practice Needs Scale (MPNS-36) questionnaire assuming a total MPNS mean score of 1.82 (SD 0.37) at baseline[22]; a two-tailed test at 5% significance level; and allowing for an intracluster correlation of 0.05% and 20% loss to follow-up between baseline and end line surveys.

In addition, we will obtain a random sample of 10 girls in each school from the participants in the female cohort who reported in the baseline questionnaire to have missed two or more days of school during and because of their last menstruation to qualitatively assess changes in school participations and experiences of the intervention. We will also purposively select two staff in each school based on individual involvement in MSRH issues within the school to participate in KIIs (see table 1 for more details).

No participants will receive monetary incentives. All will receive MSRH education and workbooks; girls will receive a kit containing, among other things, a reusable menstrual product.

## Intervention delivery and components

The intervention will be delivered by Femme after refinement during the formative phase. The evaluation will be conducted by a MITU/LSHTM team (see figure 2). The following components are expected to be part of the finalised intervention:

**Comprehensive education sessions** will be delivered by Femme facilitators immediately following regular school activities to all Form 2 and 3 girls and boys (~500 girls and ~500 boys, age range 14–18 years) in four schools. Ideally sessions will run over 5 days and last approximately 1–2 hours per session; however, this may vary by school depending on their timetable and availability. In each school, girls and boys will first be together for two sessions covering anatomy, puberty, virginity, relationships and gender roles, basics of menstruation, reproductive health and pregnancy. Two subsequent boys'-only sessions will cover hygiene, physical (sexually transmitted infections, urinary tract infections) and mental (anxiety, substance abuse) health, gender-based violence and relationships. Three girls'-only sessions will cover hygiene, menstruation (management, reusable products), pregnancy, physical health (RTIs, STIs, UTIs), anxiety and pain management. The formative phase will inform about the frequency and distance between sessions.

**Menstrual pain management** will likely include the provision of strategies such as application of heat, dietary options, mild exercise/stretching, recommendations for over-the-counter analgesics such as Panadol or Ibuprofen, and incorporate anxiety-reducing and pain-reducing

breathing techniques during the girls only sessions. During the formative phase, we will also explore (with the schools and LGAs) the best way for such analgesics to be made available and dispensed within the schools.

**Distribution of reusable menstrual products** (menstrual cups, reusable pads, period underwear). During the girls'-only sessions, students will be taught the advantages, disadvantages and usage of all products. At the end of the education sessions, girls will receive a Femme kit. If deemed appropriate during the formative phase, a boy's kit will be developed and delivered in the pilot, such a kit could include information booklets, soap in a container, underwear.

**Two post-workshop check-in sessions** will be conducted, one at 6 weeks after the education sessions (with both boys and girls) and the other at 12 weeks (with girls only). The aim of these sessions will be to reinforce key workshop messages and support girls' product uptake. These sessions are pivotal to reinforcing key learnings while tackling myths and misconceptions, answering questions and providing support for product usage, especially menstrual cups. Students will be guided through a quiz with specifically designed questions that will elicit conversation regarding the key topics, including: UTIs, menstrual pain management and vaginal discharge. In these sessions, Femme facilitators will identify early adopters of the menstrual cup, support them and build an encouraging support group to promote the use by others who also selected the product but may have not yet tried or have experienced challenges when first trying. Teachers will not be present during workshops or check-ins, as students will not feel free to ask pertinent and personal questions.

**Teacher/staff workshops** will take place separately in each of the four schools, before the student education sessions occur, with all teachers. They will cover the same topics as listed for the students, provide opportunities to ask questions, and female teachers will receive a reusable menstrual product. One of the secondary outcomes of the project is to improve school climate hence the involvement of boys and male teacher is an important strategy in the project. Both male and female teachers will participate in key informant interviews during formative and stakeholders' workshops at the beginning of the intervention.

**School wash improvements** in the four schools will be based on information collected from students, teachers and districts during the formative phase and will be implemented by relevant people within each school, in partnership with Femme facilitators prior to the first workshop. Improvements are likely to include door locks, hooks on doors, soap, bins and buckets for water.

**District and school level meetings** will take place throughout the project period at suitable time points identified during the formative phase to coincide with relevant government decision-making and budgeting time points. By aligning implementation planning with government timelines, there is opportunity for intervention inclusion in annual plans, strategic plans and budgets.[23] These meetings are intended to explore mechanisms for integration of the intervention components described above into the education system.

## Outcome evaluation

A mixed-methods evaluation will be conducted in the four intervention delivery schools (see figure 3). The study design will be a single-arm longitudinal study with pre-post survey among students, and qualitative studies among students and staff (see table 1). For the quantitative pre-post evaluation, all girls and a subsample of 50 boys per school registered in Forms 2 and 3 will be recruited. The study population for the quantitative evaluation will comprise adolescents aged 13–18 years attending the four intervention schools.

Qualitatively, we will conduct 20 sequential in-depth interviews at three time points (before, immediately post intervention, 12 months later) with five girls from each school, who reported in the baseline questionnaire, to have missed school for two or more days during and because of their last menstruation. The aim of these sequential in depth interviews (IDIs) is to assess any changes in participation and to explore if/how the intervention may have impacted on the girls' ability to participate in school. We will also conduct KIIs with two staff per school to assess MSRH knowledge among staff overseeing student's MSRH issues while in school. All interviews and FGDs will be conducted in Swahili by a trained female research assistant accompanied by a note taker and recorded using digital voice recorders.

We will assess one primary and eight secondary outcomes. The data collection tools for each outcome are described in table 2.

## Process evaluation

We will triangulate multiple data sources to examine elements of fidelity, coverage and uptake, sustainability, and acceptability as well as capturing how the social, cultural and political context has affected implementation, as described below:

### Fidelity

We will conduct (1) observations of all the government and parent meetings, spot check observations of WASH facilities pre, immediately post then 12 months post intervention, and observations of 20 Femme-led workshops sessions with girls (n=10), boys (n=8) and teachers (n=2); and (2) we will conduct unstructured informal interviews with the Femme facilitation team to assess how training went and to what degree actual implementation of each component varied from what was planned. Modifications in activities, frequency or timeline will be captured.

### Coverage and uptake

We will capture the number of participants reached through the different intervention components, stratified by student/teacher, sex and age. This information will be collected by the Femme monitoring tool and verified during evaluators' observations of a sample of the sessions.

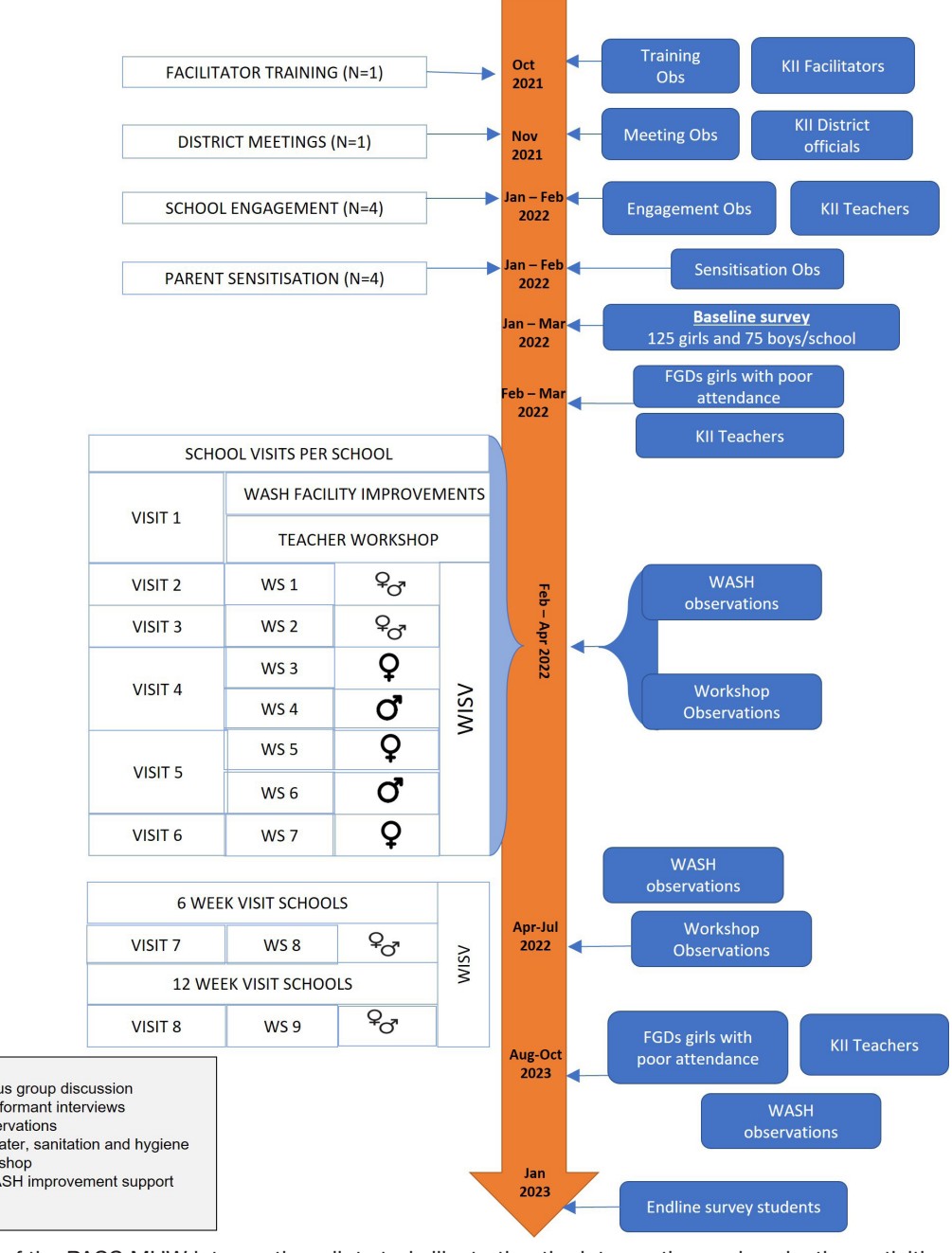

**Figure 3** Flow chart of the PASS MHW intervention pilot study illustrating the intervention and evaluation activities.

*Sustainability*
Unannounced WASH spot check observations will be conducted at two time points post intervention by the MITU team (7 and 12 months), which will be in addition to the routine monitoring and evaluation by the Femme implementation team (6 and 12 weeks). All observations of the WASH facilities will use an adapted observation checklist (see table 2).

*Acceptability*
Acceptability of the intervention will be assessed using two FGDs per school with up to 10 girls and 10 boys who received the intervention, at two time points, months 1 and 12 post intervention. The students will be randomly sampled from a list of all those who attended

the intervention sessions. Additionally, three KIIs will be conducted with teachers and two KIIs per district with district officials at these same time points, to assess the acceptability of the school intervention.

*Context*
Social, cultural, political and logistical factors that may have facilitated or inhibited the implementation of the intervention will be explored using IDI and FGD among students and teachers from school interventions at the end of the intervention at 12 months. Contextual reasons for adaptations to the intervention and its delivery will be also explored.

**Table 2**

| Outcome | Population and method* | Proposed scale to finalise in the formative phase |
|---|---|---|
| **Primary outcome** | | |
| Menstrual practices and perceptions | (1) | Menstrual Practices Needs scale.[22] 36 questions to measure perceptions of comfort, satisfaction, adequacy, reliability and worries/concerns during the last menstrual period |
| **Secondary outcome** | | |
| 1. Ability to manage pain | (1) | 3 questions have been drawn from Femme routine M&E tools and will be modified during the formative phase, but assess the presence and severity of pain during the last menstrual period. |
| 2. Self efficacy of management of menstruation | (1) | Self-efficacy in Addressing Menstrual Needs Scale (SAMNS) is 26-item scale, which measures of a schoolgirl's confidence in her capabilities to address her menstrual needs. The tool comprises 3 correlated subscales: The SAMNS comprises 3 correlated subscales: Menstrual Hygiene Preparation and Maintenance (MHPM, 17 items), Menstrual Pain Management (MPM, 5 items), and Executing Stigmatised Tasks (EST, 4 items). |
| 3. Menstrual-related anxiety | (1) | Generalised Anxiety Disorder is a seven-item scale (GAD7) which has been translated and tested in Swahili among this population |
| 4. Self-reported reproductive ill-health symptoms | (1) | 9 questions drawn from Femme routine M&E tools to assess the presence of a series of symptoms in the last month. |
| 5. MSRH Knowledge | (1), (2) | 11 multiple choice questions have been drawn from Femme routine M&E tools and MENISCUS study in Uganda to assess knowledge of puberty in general and MSRH |
| 6. School and social participation | (1), (3) | Quantitatively we have included 8 questions in the survey to assess participation. Qualitatively, the girls will describe experiences of their LMP including the days, activities, sessions missed at LMP because of menstruation. We will explore drivers for suboptimal participation and reasons for any changes over the course of the project. |
| 7. School WASH facility improvements | (1), (4) | Baseline and endline checklist to capture the presence of and access to toilets, water sources, hand-washing stations and disposal methods. Questionnaire also contains eight questions to assess the availability, access and condition of WASH facilities in the school |
| 8. School climate | (4) | Questions adopted from the Beyond Blue School Climate questionnaires.[22] The scale has 28 items in four subscales: relationships (between students and teachers), belonging, commitment and participation. |

(1) Quantitative survey of a cohort of all girls: census of all girls in Forms 2 and 3 at baseline.
(2) Quantitative survey of a cohort of 200 boys randomly selected at baseline from Forms 2 and 3.
(3) Sequential IDIs with 20 girls who (at baseline) reported to have missed two or more days of school during and because of their last menstrual period who will be interviewed at three time points.
(4) Four schools—baseline and endline assessment.
*Data sources.
WASH, water, sanitation and hygiene .

## ECONOMIC EVALUATION

A costing study will be conducted to estimate the total costs of developing, setting up and running the intervention. A combination of top-down and ingredients-based costing approaches will be used to generate cost estimates for the whole intervention, and for each component. All costs will be estimated from the societal perspective (the students, the schools and implementing partners) and financial and economic costs will be calculated for all inputs. Costs will be collected using accounting records, staff time survey and process evaluation data including interviews with participants. Capital costs and those costs with a lifespan greater than the project life will be annualised over their useful life. Costs will be entered into an Excel-based costing tool. The cost analysis will describe the distribution of costs across different forms of inputs, and will estimate the unit cost per student reached, the cost of delivering all activities in schools, and the cost per unit of measure for selected intervention outcomes. The cost estimate will be compared with similar school programmes in the region and will inform programme replication, scalability and financial sustainability for different implementing partners.

### Data management and analysis
#### Quantitative data
The quantitative survey data will be entered into customised MySQL databases and transferred to STATA V.15 (Stata Corp College Station, Texas, USA) for cleaning and analysis. Paper source documents will be stored in lockable cabinets within the data room at MITU which is locked and accessed by data mangers only. Electronic data will be stored on servers at MITU. Descriptive analysis will be conducted with baseline data using mean

(SD) or median (range) to summarise continuous variables. Similarly, frequency and percentages will be used to describe participants' characteristics collected using categorical variables. The effect of the intervention on the primary outcome of MPNS—a continuous outcome—will be assessed with random-effects linear regression model adjusting for baseline measure, and including individual as a random effect and school as a fixed effect. For binary secondary outcomes, random-effects logistic regression will be used to estimate adjusted ORs with random effects for individuals to account for within-individual clustering of baseline and end line responses.

### Qualitative data

Detailed field notes will be taken by a research assistant; additionally FGDs, KIIs and IDIs will be recorded using digital voice recorders. Immediately post data collection the research assistant and study coordinator will use the field notes and recordings to prepare detailed field notes. Additional post data collection debrief meetings will take place after every 2 days of data collection between the data collection team (RA and study coordinator) and one of the study principal investigators. Immediately post data collection the research assistant and study coordinator will use the field notes, recordings and debrief meetings to prepare analytical memos at the individual interview/FGD and then school level. The recordings will then be transcribed then translated and stored in an interoperable document file format (*.odt) recommended for long term preservation. A thematic analysis approach will be used, which will include the following steps: (1) grouping of the statements and words according to similarity in meaning; (2) we will then apply codes (meaningful labels) to the groupings; and then (3) we will group the codes into categories (codes), which are eventually grouped into themes. We will employ grounded theory principles of constant comparison, theoretical sampling and saturation, and analytical memo writing during data collection and analysis. NVivo shall be used during data analysis. We will maintain rigour through ongoing discussion within the research team regarding each step of the research process and the resultant interpretations from the data. We will use verbatim quotes to support our interpretations of the data and pay close attention to disconfirming cases and divergent views of the participants. Lastly, we will present preliminary study findings to the intervention team and the district facilitation team to discuss and reach consensus on the interpretation and the way in which we have represented the study findings.

### Ethics and dissemination

All participants will provide written informed consent. Head teachers will provide overall consent on behalf of adolescents aged below the age of 18 years before they give assent. Each participant will be informed that participation in the study is voluntary and that they are free to withdraw, without justification, from the investigation at any time without consequences. Information about the aims of the study and its methods will be provided to the participants before asking for their written informed consent or assent to participate in an interview or FGD. If the respondent has agreed for the interview to be recorded, then the consent procedure will be digitally recorded using handheld devices. Each respondent will be assured of the confidentiality and privacy during data collection, management and analysis of data. All interviews will be conducted in private, out of ear shot of other persons. All information from the respondents will be kept confidential. Personal data will be anonymised using ID numbers, and stored data will be stripped of names, and password protected for use by named research staff only.

We envisage minimal risk should a girl be 'seen with a Femme Kit'. Our project aims to demystify and de-stigmatise menstruation. In partnership with multiple stakeholders (teachers, parents, boy and girl students), project activities include developing school-level action plans to create a supportive environment. All participants will benefit from increased knowledge and understanding of both their own body and how it works, as well as that of the opposite sex. Additionally, female teachers and students will benefit from improved menstrual management strategies (including pain reduction, anxiety reduction) and a reusable menstrual product. The intervention is hypothesised to reduce menstrual-related stigma through provision of education and stakeholder engagement. All participants will be given information about local SRH services at the end of their interview. Precautions will be taken to control and modify potential discomfort, distress or hazards to research participants.

Participants will not be paid for taking part in the study. The only payments made to participants will be to compensate for travel costs for those who will have to travel in order to participate in the study. Participants who will take part in the stakeholders' meetings and FGDs will be given refreshments during the meetings/FGDs.

Protocol, tools and consent have been reviewed and approved by the independent Tanzanian national ethics committee (Ref: NIMR/HQ/R.8a/Vol.IX/3647) and LSHTM ethical review board (LSHTM Ethics Ref: 22854).

Our dissemination plan allows for flexibility to be responsive to any contextual changes. Our targeted audiences are both academics and non-academic audiences. Our dissemination plan will form part of the overall pathway to impact plan for the project. Local and national stakeholders will be involved throughout the life of the project starting with involvement during the refinement of the intervention. Study findings will be disseminated to local stakeholders through presentation in relevant stakeholders' meetings such as the Tanzania Menstrual Health Day, and sharing key findings through webinars and social media. We will also share study findings in relevant international conferences and publish in open-access peer-reviewed journals.

## Current status

At the time of publication, the formative phase and preliminary government engagement will be finished. Phase III implementation begins in January 2022 and ends in January 2023.

### Author affiliations
[1]Mwanza Intervention Trials Unit, National Institute for Medical Research, Mwanza Research Centre, Mwanza, Tanzania
[2]Monitoring and Evaluation, Femme International, Moshi, Tanzania
[3]Disease Control, London School of Hygiene and Tropical Medicine, London, UK
[4]Infectious Disease Epidemiology, London School of Hygiene and Tropical Medicine, London, UK
[5]Global Health and Development, London School of Hygiene and Tropical Medicine, London, UK
[6]Epidemiology and Biostatistics, Kilimanjaro Christian Medical University College, Moshi, Tanzania
[7]Population Health, London School of Hygiene and Tropical Medicine, London, UK

**Acknowledgements** We would like to thank the previous participants of Femme International for their inputs. Through their involvement, Femme International was able to identify gaps in their programming that this PASS study aims to address.

**Contributors** EO, J Rubli and J Renju conceived the study concept, developed the funding proposal, applied for funding and initiated the writing of the protocol paper. BT, SK, GG and KM provided critical review of the content. PA contributed to designing the quantitative component, sample size calculation and developed the statistical analysis plan. All authors revised and approved the final version of the manuscript.

**Funding** This work was supported by the Medical Research Council (MRC), UK Grant Ref: MR/T040297/1.

**Competing interests** None declared.

**Patient and public involvement** Patients and/or the public were not involved in the design, or conduct, or reporting or dissemination plans of this research.

**Patient consent for publication** Not applicable.

**Provenance and peer review** Not commissioned; externally peer reviewed.

**ORCID iDs**
Jennifer Rubli http://orcid.org/0000-0003-1657-8133
Jenny Renju http://orcid.org/0000-0001-5650-1902

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
