## [Reviewer comments · BMJ Open]

ARTICLE DETAILS

TITLE (PROVISIONAL)	The co-development and piloting of a menstrual, sexual and reproductive health intervention to improve social and psychological outcomes among secondary school girls in Northern Tanzania: The PASS MHW study protocol
AUTHORS	Okello, Elialilia; Rubli, Jennifer; Torondel, Belen; Makata, Kenneth; Ayieko, Philip; Kapiga, Saidi; Greco, Giulia; Renju, Jenny

VERSION 1 – REVIEW

REVIEWER	Sebert Kuhlmann, Anne Saint Louis University
REVIEW RETURNED	12-Oct-2021

GENERAL COMMENTS	The co-development and piloting of a menstrual, sexual and reproductive health intervention to improve social and psychological outcomes among secondary school girls in Northern Tanzania: The PASS MH study protocol. This protocol lays out an interesting and important research study designed to develop an effective menstrual hygiene intervention for secondary school students in northern Tanzania. Overall, it appears to be a well-planned study. The protocol presentation could use some clarification, however, to improve its readability. In the Introduction, it would help to provide existing school drop-out and attendance rates broken down by gender, not just overall. Recently, we have seen in many places that absences and drop-outs are actually higher among male students even without or prior to menstrual hygiene interventions for female students, so having such a background gender breakdown will allow the reader to interpret better the results. The Methodology could use several clarifications. For example, are both male and female teachers being recruited into the study? What is the justification for not providing incentives for participation? What is the justification for the selected sample sizes? For example, what suggests that 2 key informant interviews with staff and 2 focus groups by gender will be sufficient to reach saturation and inform the study properly? It would also help to provide a schematic or visual representation of the proposed study,
------------------	--

	as it is complicated. While Figure 2 presents some of this, it is not categorized by Phase and Step or Location the way it is described in the text. This makes it difficult to follow, hard to connect to the text, and unclear about what components are part of which phases, steps, and locations. Recommendation: revise and resubmit
--	--

VERSION 1 – AUTHOR RESPONSE

Reviewer: 1

Dr. Anne Sebert Kuhlmann, Saint Louis University

Comments to the Author:

7. This protocol lays out an interesting and important research study designed to develop an effective menstrual hygiene intervention for secondary school students in northern Tanzania. Overall, it appears to be a well-planned study.

We thank the reviewer for this comment and for their recognition of the value of the study

8. The protocol presentation could use some clarification, however, to improve its readability. In the Introduction, it would help to provide existing school drop-out and attendance rates broken down by gender, not just overall. Recently, we have seen in many places that absences and drop-outs are actually higher among male students even without or prior to menstrual hygiene interventions for female students, so having such a background gender breakdown will allow the reader to interpret better the results.

This has been revised, we have included recent school drop out data by sex on page 7 lines 2-5

9. The Methodology could use several clarifications. For example, are both male and female teachers being recruited into the study?

Yes both male and female teachers are included in the different stages of the study, we recognise the need to hear from both male and female teachers, in the formative phase we state that we will conduct FGDs stratified by gender Page 12 line 24. In step 2 (of phase 1) we have now further clarified (line 9 page 13) that we will include both male and female teachers. Finally in phase 2 (implementation of the pilot intervention) we have included more details on the involvement of both male and female teachers, see page 21 lines 20-23.

10. What is the justification for not providing incentives for participation?

On page 14 line 20 we explain what the incentives/motivations for participation, namely education, information booklets and the provision of menstrual health materials to girls. From the past experience of Femme these actions/items were a powerful incentive/motivation to participation.

11. What is the justification for the selected sample sizes? For example, what suggests that 2 key informant interviews with staff and 2 focus groups by gender will be sufficient to reach saturation and inform the study properly?

We recognise reviewers concern about saturation. However, in the qualitative component of the study we have triangulated different methods of data collections to include KII, Observation and Focus group discussions. We are also triangulating data from different study participants to include students, teachers and other key stakeholders. The numbers presented are per category of participants and collectively the data from the different methods and participants capture a broad perspective of views. Another point of consideration is that for some participants e.g. head teachers, District education officers and Guidance teachers (those focused on health education) the total sample available is

limited (e.g. one per school or one per district), in this small pilot study it is therefore not possible to increase this sample size. Finally we will be cognisant of the limitations of these methods in our analysis and interpretation.

It would also help to provide a schematic or visual representation of the proposed study, as it is complicated. While Figure 2 presents some of this, it is not categorized by Phase and Step or Location the way it is described in the text. This makes it difficult to follow, hard to connect to the text, and unclear about what components are part of which phases, steps, and locations.

Many thanks for this suggestion, we have now added a figure to the manuscript – cited in the text page 12, line 1

Yours faithfully,

Jenny Renju (on behalf of the co-authors)

VERSION 2 – REVIEW

REVIEWER	Sebert Kuhlmann, Anne Saint Louis University
REVIEW RETURNED	03-Dec-2021
GENERAL COMMENTS	This revision is now acceptable for publication. It could still use a thorough review to check for typos, etc. In addition, it would also be help to include how the sample size estimates were calculated for the Phase 2 quantitative survey. Note that the figures are illegible in this version, even when zooming it.

VERSION 2 – AUTHOR RESPONSE

Reviewer: 1

Dr. Anne Sebert Kuhlmann, Saint Louis University

Comments to the Author:

This revision is now acceptable for publication. It could still use a thorough review to check for typos, etc.

Response: Many thanks we have now done this throughout the whole document

In addition, it would also be help to include how the sample size estimates were calculated for the Phase 2 quantitative survey.

Response: Page 13 line 19-24 - we have included more details in the revised manuscript to describe how the sample size estimates were calculated: "We determined that 500 girls - each participating in the baseline and end line survey - were required to provide 80% power to detect a difference of 0.075 points in the Menstrual Practice Needs Scale (MPNS-36) questionnaire assuming a total MPNS mean score of 1.82 (SD 0.37) at baseline a two-tailed test at 5% significance level; and allowing for an intra-cluster correlation of 0.05 and 20% loss to follow up between baseline and end line surveys".

Note that the figures are illegible in this version, even when zooming it.

Response: We have modified the figures and now they are suitable for publication,